# Insect Protein as a Component of Meat Analogue Burger

**DOI:** 10.3390/foods13121806

**Published:** 2024-06-08

**Authors:** Anna Krawczyk, Juana Fernández-López, Anna Zimoch-Korzycka

**Affiliations:** 1Department of Functional Food Products Development, Faculty of Food Science and Biotechnology, Wrocław University of Environmental and Life Sciences, 37 Chelmonskiego Str., 51-630 Wrocław, Poland; anna.zimoch-korzycka@upwr.edu.pl; 2IPOA Research Group, Institute for Agri-Food and Agri-Environmental Research and Innovation Miguel Hernández University (CIAGRO-UMH), Ctra. Beniel km 3.2, 03312 Orihuela, Spain; j.fernandez@umh.es

**Keywords:** edible insect, *Alphitobius diaperinus*, alternative protein source, consumer acceptance, quality properties, texture profile

## Abstract

Researchers are exploring solutions to meet the growing demand for protein due to the expected increase in global population by 2050. Interest in alternative protein sources like insects has risen, driven by concerns about environmental impact and the need for sustainable food production. This study aimed to develop and evaluate the physicochemical properties of soy-protein-based burgers enriched with insect protein from *Alphitobius diaperinus*. Three formulations were developed: a control (B0) and burgers with 5% (B5) and 10% (B10) insect protein—Whole Buffalo Powder (WBP). The results showed that adding insect protein decreased the burger analogue’s pH. A clear trend was observed of increasing total lipids and saturated fatty acids (SFA) and decreasing monounsaturated (MUFA) and polyunsaturated fatty acids (PUFA) as the WBP concentration increased from 0% to 10%. No significant differences with increasing WBP concentration in the protein content of the burger analogue, as well as the cooking yield, were noted. The WBP addition had a notable effect on the color change, especially a decrease in brightness (L*). It was shown that as the WBP concentration increased, there were no significant differences in the texture profile of the burger analogues. The formulation with 5% WBP concentration was the most acceptable in sensory analysis.

## 1. Introduction

The United Nations predicts that the global population may reach 9.7 billion by 2050 [1], but meat production can only meet the needs of nearly eight billion people. This significant growth in population is expected to result in a surge in the demand for animal-protein sources, which would force the meat industry to increase production by about 50–73% in order to meet the daily requirements of the expanding population. Malnutrition, which affects vulnerable groups such as young children, the elderly, and those with compromised immune systems, exacerbates the challenge of meeting nutritional needs globally. Apart from insect protein, there is no other sources of high-quality proteins with large-scale production. Therefore, future sustainable development may face challenges [2,3]. The availability of finite resources, such as farmland and freshwater, to meet the food needs of the growing population is a growing concern [4].

Sustainable food production with low environmental impact has become a crucial issue. The livestock industry has been unsustainable and has contributed to climate change, responsible for 14.5% of global greenhouse gas emissions and consuming up to 30% of freshwater resources [5,6]. Establishing new farms has been linked to deforestation, pollution, damage to hydrogeological reserves, and threats to biodiversity [7]. Continuing to rely on the livestock sector to meet our meat or protein needs will have many adverse environmental impacts. Therefore, plant analogues have witnessed a surge in popularity and their market is growing [8,9]. However, despite the enthusiasm surrounding plant-based analogues, it is important to realize that they may not be as ideal as is commonly believed. While these alternatives are often advertised as healthier options, they can vary significantly in nutritional composition, with some formulations lacking essential nutrients such as essential amino acids, vitamins (e.g., B12), and minerals (e.g., iron). Studies also indicate that the digestibility of plant-derived proteins is much lower than those of animal origin [10,11,12,13].

Plant proteins are commonly viewed as a sustainable protein source; however, their production system is not without its drawbacks. The primary concern revolves around monoculture farming, where the same plant species is cultivated in the same field for multiple years, leading to adverse impacts on biodiversity and soil fertility. Moreover, there is growing concern regarding the negative effects of pesticides and chemical fertilizers, which can seep into food and accumulate in plant cells. Escalating climate change is causing more frequent droughts during the growing season, resulting in a significant decrease in both the quantity and quality of crop yields. Consequently, current plant protein production may not be adequate to meet future protein demands [14,15,16].

The challenges posed by climate change and continuous population growth have led to the search for alternative protein sources for humans, such as insects, fungi, cultured meat, and micro- and macroalgae, which are nutritionally healthy and can be obtained more efficiently and sustainably than traditional sources of protein [17]. Therefore, edible insects possess the potential to be incorporated within a global strategy aimed at attaining food security on a global scale. Insects represent a substantial and diverse living resource on our planet, boasting an impressive count of 5.5 million species. Among this vast array, nearly 2000 insect species are actively consumed across 113 countries, predominantly in Africa, South America, and Southeast Asia, where the practice of consuming insects, known as entomophagy, holds deep historical roots, dating back at least 3000 years [18].

The consumption of insects is often considered ‘disgusting’ in the Western world, with the unwillingness to introduce them into the diet defined by the term neophobia [19,20]. Food neophobia a the term used to describe the fear or dislike of trying new food items, leading to a reduced willingness to include them in one’s diet. This aversion is influenced by individual traits, cultural aspects, and socioeconomic conditions, which can restrict exposure to unfamiliar foods [21,22]. Obstacles such as unfamiliarity, sensory variations, and inherent disgust and fear of new things present significant challenges to achieving broader acceptance of edible insects [23,24]. Although, numerous research studies have been conducted on the significance of edible insects as a substitute for other sources of protein. The primary emphasis in edible insect market research has been on consumer behavior and attitudes. Despite the reluctance of many consumers towards this type of new food, market research indicates an increase in the number of producers and consumers. The global edible insects market is projected to reach a forecast value of around USD 5.5 billion by 2026, exhibiting a compound annual growth rate (CAGR) of 33.72% [25]. The insect protein market in Europe is led by the Netherlands, France, Germany, UK, and Belgium, driven by a combination of regulatory support, innovation, and economic viability. These countries have also been proactive in addressing consumer perception and market opportunities, implementing initiatives to educate consumers about the benefits of insect protein and integrating insect-based products into mainstream markets [26,27,28]. Insects exhibit several advantageous traits that make them a promising candidate for sustainable food production. Firstly, they possess high fecundity rates and can breed year round. Secondly, they have high conversion rates, which means they can efficiently convert feed into body mass; for instance, crickets require one-twelfth the amount of feed compared to cattle, one-fourth that of sheep, and half as much as pigs and broiler chickens to yield an equivalent amount of protein. Moreover, insects have a low environmental impact, primarily due to their low greenhouse gas emissions, and require minimal breeding space. Lastly, certain insect species have the ability to recycle organic industrial and agricultural byproducts, which can be used as a source of feed for livestock or humans [19,29]. Consuming insects has the potential to alleviate animal suffering in comparison to the consumption of conventional livestock. In addition to the aforementioned environmental advantages, insects possess significant nutritional value, as they are notably abundant in high-quality protein that consist of crucial amino acids, such as tryptophan lysine, tryptophan, and threonine [30,31].

Within the European Union, the use of edible insects for food production is subject to stringent regulatory measures established by the European Food Safety Authority (EFSA) and national authorities. In the current regulatory landscape, the EU Novel Food Regulation (EU) 2015/2283 (entered into force in 2018) governs the authorization and marketing of novel foods, including edible insects [32,33]. Notably, certain insect species have been evaluated and approved under this regulation for use in food and feed applications. These species have undergone comprehensive safety assessments to ensure their suitability for human consumption, considering factors such as allergenicity, toxicological properties, and nutritional composition. Four species of edible insects have been officially approved, including the yellow mealworm (*Tenebrio molitor*), migratory locust (*Locusta migratoria*), house cricket (*Acheta domesticus*), and lesser mealworm (*Alphitobius diaperinus*) [34,35]. Among the listed species, *Alphitobius diaperinus* has been authorized by EU authorities relatively recently and is identified as the one with the greatest potential for use as food and feed in the EU [36,37]. *Alphitobius diaperinus* possesses a greater protein concentration in dry matter (~64%) in comparison to the previously mentioned insects [36,38,39,40,41]. This makes it a competitive protein source comparable to traditional animal proteins. Additionally, the carbohydrate content is approximately 21.8% of dry matter (DM), providing a substantial energy source. The lipid content varies significantly, ranging from 13.4% to 29.0% DM, which highlights its potential as a source of essential fatty acids. The ash content, which indicates the total mineral content, is reported to be 3.6% DM, reflecting the presence of various essential minerals. In terms of specific mineral content, *A. diaperinus* contains 0.5 g/kg DM of calcium, 21.9 mg/kg DM of copper, 53.5 mg/kg DM of iron, 10.0 g/kg DM of potassium, 1.3 g/kg DM of magnesium, and 5.4 mg/kg DM of manganese. These minerals are crucial for various physiological functions, making this insect a nutritionally dense food source. The high iron content, in particular, is beneficial for addressing iron deficiency, while the presence of other minerals supports overall metabolic health [36,42,43,44]. Moreover, *A. diaperinus* exhibits an accelerated developmental cycle and enhanced reproductive capacity, leading to decreased production costs per unit mass. Consequently, it emerges as a financially accessible and nutritionally advantageous choice for consumers, serving both as a source of food and feed [45].

Despite the high nutritional value of *Alphitiobus diaperinus*, the threats posed by its allergenicity and the microbiological quality must be taken into consideration when assessing its suitability as a novel food. Insect allergies, though rare, can still affect susceptible individuals. The risk of allergic reactions is linked to specific proteins that act as allergens. Researchers have identified tropomyosin as a common allergen in lesser mealworms and have also found peptides similar to known allergens like arginine kinase. This suggests that people sensitive to house dust mites and crustaceans may experience cross-reactivity with these insect allergens. Additionally, it has been found that individuals allergic to prawns are at a higher risk of developing food allergies to mealworms and other insects [37,43]. Studies have shown that the microbiome associated with *A. diaperinus* can vary significantly depending on environmental conditions and storage methods [46]. The safety of frozen and freeze-dried formulations of A. diaperinus has been evaluated by the European Food Safety Authority (EFSA), which noted that the levels of contaminants in these formulations depend largely on the insect feed [37]. Additionally, long-term storage studies have found that while most stored insect materials are safe for consumption, there have been instances where microbial limits were exceeded, emphasizing the need for stringent monitoring and proper storage conditions [47]. Therefore, while A. diaperinus offers substantial nutritional benefits, its allergenic potential and microbiological risks must be carefully managed through proper processing, handling, and regulatory oversight to ensure consumer safety. Insect proteins are being researched for their usefulness, for example, as new food ingredients to increase the protein content of foods, to replace animal proteins, to and enrich food products with essential amino acids, with positive effects on nutritional value [48]. The idea of creating a plant burger with the addition of insect protein is intended to accustom consumers to the specific taste and aroma of this component, especially in the Western world, where consumers are not accustomed to it [19]. Therefore, the aim of the present study was to design and develop a burger-type meat analogue using insect protein (*Alphitobius diaperinus*) in different concentrations and to evaluate its effect on the physicochemical properties of the product, including pH, protein content, cooking yield, texture profile analysis (TPA), color, and sensory acceptability, in comparison to a soy, plant-based burger.

## 2. Materials and Methods

### 2.1. Materials

Fiber Textured Insect Protein (FTIP) (12 g protein, 9 g lipid, and 70 g H_2_O/100 g fresh weight) and Whole Buffalo Powder (WBP) (57 g protein, 27.6 g lipid, and 5 g H_2_O/100 g fresh weight) of lesser mealworms (*Alphitobius diaperinus*) were obtained from Protifarm Processing B.V. (Ermelo, The Netherlands). In addition, soya chops (49 g protein, 1.4 g lipid, 20 g carbohydrates, and 16 g of fiber) and soya isolate with a protein content of 99.9% were purchased from Sante Sp. z o.o. (Warsaw, Poland and Novichem Sp. z o.o. (Chorzów, Poland), respectively. The components of meat analogue burger used in this study are shown in Table 1.

### 2.2. Methods

#### 2.2.1. Variants Preparation

Three formulations of the product were prepared, based on the soya chop: B0—control, without added insect protein; B5—with 5% WBP content; and B10—with 10% WBP content. In order to introduce mealworm protein into the product, in variants B5 and B10, the amount of soybean chop and soy protein isolate was partially substituted with FTIP and WBP in the amount of 2.85 g and 2.59 g, respectively, for trial B5, while variant B10 was substituted with 5.7 g and 2.59 g, respectively. Therefore the protein content of the samples was at the same level compared to the control sample, B0. The criteria used to select such concentrations in experimental variants was preliminary sensory analysis. The burgers compositions are outlined in Table 1.

The initial stage of the burger analogue preparation consisted of soaking the soy chops in hot water, in a ratio 4:6 (soy:water)—for variant B0, 20.4 g of water was used, while in B5 and B10 this amount was reduced by half. After 30 min, the chops were minced using a Diana 886.5 meat mincer (Zelmer, Rzeszów, Poland). Using homogenizer T 25 easy clean digital ULTRA-TURRAX^®^ (IKA, Staufen, Germany), an emulsion was produced (separately for each variant), consisting of the remaining water, sunflower oil as the primary oil phase in the emulsion, sodium alginate as a stabilizer and thickening agent, transglutaminase as an enzyme to improve texture and proteins cross linking, spices for flavor enhancement, and beetroot juice concentrate for natural coloring (10,000 rpm for 5 min). The coconut oil, used as a structuring agent during heat treatment and a source of fat for grilling, was previously frozen at −18 °C; using a grater, chips of about 5 mm in diameter were created. The ingredients were weighed using a PS 1200/C/1 balance (Radwag, Radom, Poland), following the previously defined additions of soy protein isolate, FTIP, and WBP. The ingredients were mixed and 50 g burgers (approx.), measuring 6 cm in diameter and 1.5 cm high, were formed and wrapped in cling film. After a 24 h incubation period at 4 °C, the product was grilled on a preheated pan to 160 ± 5 °C (Tefal S.A.S., Rumilly, France) on both sides until a temperature of 72 °C was reached at the geometric center; these measurements were conducted using a C-370 RTD Thermometer (REED Instruments, Wilmington, NC, USA), with a temperature range from −100 to 300 °C, by placing the sensor probe at the geometric center of the sample.

#### 2.2.2. Chemical Properties

##### pH

The pH of the variants was measured using a S40 SevenMulti™ pH meter (Mettler Toledo, Greifensee, Switzerland), which was calibrated using buffer solutions (pH 4 and 7) prior to analysis. Samples for measurement were prepared by mixing 10 g of raw material with 50 mL of distilled water in a beaker. Measurement was performed after 10 min of incubation at room temperature until the value on the pH meter stabilized.

##### Crude Protein Content

The Kjeldahl method, according to AN-5511, was used to analyze the crude protein content of the variants. The samples were mineralized by heating in a TecatorTM Digestor 2520 oven (FOSS, Hillerod, Denmark) to 420 °C, with the addition of concentrated H_2_SO_4_ and the catalysts K_2_SO_4_ and CuSO_4_ × 5H_2_O. Alkalization and distillation were conducted in a KjeltecTM 9 Analyser (FOSS, Hillerod, Denmark) and the total nitrogen content was converted to protein using the conversion factor N × 6.25.

##### Fatty Acids Profile

The extraction of lipids from the variants was performed using the Folch method with a mixture of chloroform and methanol in a volume ratio of 2:1. The solution was filtered and the supernatant was evaporated to dryness using a Rotavapor^®^ R-215 vacuum evaporator (Büchi, Flawil, Switzerland). After lipids extraction, the fatty acid profiles were analyzed.

A total of 50 mg ± 1 mg of each extracted sample was placed in a hydrolysis tube with the addition of a few boiling stones, 4 mL of 0.5 M NaOH in MeOH, and the same amount of 14% BF_3_ in MeOH. The sealed tubes were placed in a water bath (70 °C for 30 min). After the time had elapsed, the tubes were cooled in ice water and 1 mL of saturated NaCl solution was added to the hydrolysate. Then, the fatty acid methyl esters were extracted three times with 2 mL of hexane. The combined hexane layers were dried through a layer of anhydrous magnesium sulphate, evaporated to dryness under reduced pressure (50 °C; 150 mbar pressure) on a Rotavapor^®^ R-215 vacuum evaporator (Büchi, Flawil, Switzerland), resuspended in 1.5 mL hexane, and subjected to chromatographic analysis. Fatty acids were determined after methanolysis (0.5 M NaOH/MeOH and 14% BF3/MeOH) via gas chromatography combined with a mass spectrometry (GC/MS) technique using a GC6890/5973 MSD instrument (Agilent Technologies, Inc., Santa Clara, CA, USA). An HP88 column (length: 100 m, diameter: 0.25 mm, and stationary phase film thickness: 0.20 μm) was used. The carrier gas was helium 6.0 purity (Air Products, Siewierz, Poland), the flow rate of which was set at 1 mL/min; the sample injection was performed using a split (split 4:1), and the temperature program was followed using a temperature ramp: initial temperature 60 °C maintained for 2 min; heating 20 °C/min to 180 °C; 3 °C/min to 220 °C maintained for 15 min; and final heating at 5 °C/min to 250 °C, with this temperature maintained for 8 min. The total analysis time was 50.33 min [49]. Fatty acids were identified through the comparison of their retention times with standards.

#### 2.2.3. Physical Properties

##### Cooking Yield

The prepared variants were weighed and then subjected to thermal treatment, as described in Section 2.2.1. After cooling to room temperature and weighing again, the cooking yield was calculated using the following formula:$$Cooking\;yield=\frac{m_{2}}{m_{1}}\cdot 100\%$$
where: *m*_1_—mass of raw sample; *m*_2_—mass of grilled sample.

##### Color

The color value of the burger analogues was determined using a hand-held Chroma CR-400 m (Konica Minolta Sensing, Inc., Osaka, Japan). The color was recorded in the CIE-L*a*b* color space*, as defined by the Commission Internationale de l’Éclairage (CIE) in 1976, where L* is the brightness coordinate, ranging from 0 to 100 (black to white), +a*/−a* represents redness or greenness, and +b*/−b* indicates yellowness or blueness [50]. Using a white ceramic calibration plate, the instrument was calibrated (White Calibration Plate CR-A33a, Konica Minolta, Osaka, Japan) to the following coefficients values: Y = 93.5, x = 0.3114, and y = 0.319. The measurement area was 8 mm in diameter. The analysis was performed before and after heat treatment, measuring the value at three random locations on the sample surface.

##### Texture Profile Analysis (TPA)

Texture profile analysis (TPA) was conducted using a Z010 testing machine equipped with an Xforce HP load cell with a nominal force of 100 N (Zwick Roell, Ulm, Germany). The textural properties, in terms of hardness [N], cohesiveness [-], springiness [-], gumminess [N], and chewiness [N × mm], were determined using the TPA method [51]. Samples were compressed twice to a deformation of 75% with a relaxation time of 30 s. Three samples from each variant, with a cylindrical shape (15 mm × 15 mm, H × d), were prepared for this purpose and placed between two parallel plates. The analysis was performed at room temperature.

#### 2.2.4. Sensory Evaluation

The sensory analysis was carried out at Miguel Hernández University of Elche (UMH), The Polytechnic School of Orihuela (EPSO) (Alicante, Spain). To carry out the sensory analysis, a questionnaire was prepared in advance using Google Form (Google, Mountain View, CA, USA). A hedonic scale was used to assess the acceptability of burger analogues enriched with insect protein. The scale ranged from 1 to 9, where 9 meant ‘very much like’ and 1 meant ‘very much dislike’. Twenty-four panelists (*n* = 24, 50% female, 50% male, aged 22–62 years)—students and staff from the University—took part in the evaluation, and rated the product according to eight attributes: appearance, color, aroma, firmness, juiciness, taste, aftertaste, and overall acceptability. Participants were informed of the type of product tested and the allergens present. To ensure an objective evaluation, variants were blindly coded with random three-digit numbers and served on a single plate divided into three sections.

#### 2.2.5. Statistical Analysis

All analyses were performed in triplicate, unless the description indicates otherwise. The results were reported as means ± standard deviations of the measurements. The data were analyzed using one-way ANOVA analysis of variance, followed by Duncan’s post hoc test at a significance level of *p* ≤ 0.05 to test for differences between mean values. The data were analyzed using R software, version 4.3.2 (R Foundation for Statistical Computing, Vienna, Austria).

## 3. Results and Discussion

### 3.1. Chemical Properties

The burger analogues made with different concentration of WBP are shown in Figure 1. The results presented in Table 2 describes the chemical properties—pH and protein content—of variants with different WBP concentrations. The pH of all variants is slightly acidic, ranging from 6.34 to 6.78. Variant B0 has the highest pH, while B5 and B10 have lower pH values with no significant difference between them; this indicates that the addition of WBP lowers the pH of the burger analogues. A comparable relationship was shown in a study by Kim et al. [52], where the effect of the addition of edible insect protein on the physicochemical properties of a drying-induced, restructured jerky analogue was investigated—as the ratio of textured vegetable protein to insect protein decreased, the pH decreased.

The control variant (B0) had the highest protein content, while the variants with 5% WBP (B5) and 10% WBP (B10) had lower protein contents (18.41 ± 1.08% and 17.99 ± 0.24%, respectively). This decrease in protein content with increasing WBP concentration was statistically significant, which would indicate that it is concentration-dependent. The results suggest that the inclusion of WBP in burger analogues reduces the overall protein content, due to the lower protein content of the insect powder compared to the other ingredients used in the formulation. These results indicate that changing from soy protein to edible insect protein indeed reduced the overall protein content of the product. This could be related to the higher fat content of the insect protein powder [53] but also to the denaturation and aggregation states of protein that occurred during preparation [54]. Therefore, the lower protein content may be the result of higher protein leakage during thermal treatment in the B10 variant. Despite the observed decrease in protein content with higher WBP concentration, it still holds great potential as a sustainable and nutritious ingredient for burger alternatives; this is primarily attributed to its elevated levels of essential amino acids [55,56].

The fatty acid profile of burger analogues was significantly affected (*p* ≤ 0.05) by WBP concentration (Table 3). Analysis revealed that the total lipid content varied across the variants, with concentrations ranging from 31.00% ± 0.20 in the control (B0) to 35.26% ± 0.14 in the variant containing 10% WBP (B10). As the concentration of WBP increased, a notable increase in saturated fatty acids (SFA) was also observed, with the highest concentration (10%) exhibiting the highest SFA content (21.28 ± 0.004 g/100 g). Many insect proteins have higher contents of saturated fats than other sources such as plant and meat. The nutritional content of insects depends of their stage of life, habitat, and diet [57]. Monounsaturated fatty acids (MUFAs) decreased with increasing WBP concentration, with the lowest MUFA content observed in the 10% WBP variant (5.24 ± 0.003 g/100 g). Polyunsaturated fatty acids (PUFAs) also showed a decreasing trend with increasing WBP concentration, reaching the lowest level in the 10% WBP variant (8.74 ± 0.003 g/100 g). The high fat and SFA content of the samples is related to the use of sunflower and coconut oil in the formulation, and the increase in lipid content in variants enriched with insect protein powder is attributed to the higher fat content naturally present in *Alphitobius diaperinus* in comparison to texturized soy [58,59]. In animal organisms, such as insects, the content of saturated fatty acids is higher compared to plants [60,61]. In all formulations, the MUFA content was lower than the PUFA content and this is consistent when comparing the fatty acid profile of soybean and *Alphitobius diaperinus* [36,62].

### 3.2. Physical Properties

The data show a slight decrease in cooking yield as the concentration of WBP increases, but the differences between the variants were not statistically significant, as can be seen in Figure 2 (*p* ≤ 0.05). Research reported by Çabuk and Yılmaz [63] showed that the addition of insect protein powder had a negligible effect on the cooking yield of pasta compared to the control. Small differences were also noticeable when compared to pasta enriched with vegetable protein. The lower yield may be related to the higher fat content of the insect protein powders compared to the plant protein isolates [64]. The functional properties of protein such as water-holding capacity, emulsification, gelling ability, and water solubility depend on the sources, production parameters of commercial preparations, and their purity in the yield, protein and lipid contents in the final product. Due to that fact, the functional properties will vary when using different protein preparations [54].

The addition of WBP had a significant (*p* ≤ 0.05) impact on the color parameters of the burger analogues, as indicated in Table 4. Regardless of whether the variants were subjected to heat treatment or not, an increase in the concentration of WBP led to a decrease in the values of all color coordinates. Moreover, when comparing raw and grilled samples, it was observed that there was a clear tendency for the L* (lightness) and b* (yellowness) values to decrease, while the a* (redness) values increased. Alterations in the microstructure and composition in the meat analogues during cooking could affect light scattering and absorption and so would be responsible for their color changes. Several studies have assessed the color of products enriched with insect protein. Wendin et al. [65] did not explicitly describe their findings, but the figures in their paper suggest that the addition of insect flour significantly influenced the color change in the products. Furthermore, other studies directly attributed the color change in the product to the addition of insect protein, particularly noting a reduction in L* (brightness) [39,66,67,68]. This is related to the fact that insect protein preparations are not isolates of insect protein but are instead dehydrated whole ground insects and consist of other incest components. The heat treatment process of the insects increases the activity of the enzyme phenoloxidase, which catalyzes the darkening process of the insect flour [69]. In addition, studies have shown that chitin, a component of insect shells, also affects the darker color of insect powders [70].

Table 5 presents the results obtained for the TPA—hardness, cohesiveness, springiness, gumminess, and chewiness—of burger analogues. The addition of WBP had a significant effect on the cohesiveness, springiness, and chewiness of burger analogues but not on hardness or gumminess. Hardness is defined as the maximum force of the first compression cycle to a specific deformation [71]. Increasing the concentration of insect protein powder in the samples slightly decreased the hardness when compared to the control (B0). As mentioned earlier, the insect protein preparation has a higher fat content compared to the soy protein isolate, which may affect the hardness of the variants in which WBP was incorporated. Furthermore, several studies have shown that the decrease in hardness is caused by a reduction in texturization due to the addition of flour from mealworm larvae into the extruded meat analogue, which led to a weakening of the internal molecular bonds [72,73,74]. Cohesiveness is the mechanical characteristic associated with the amount of deformation that a food can undergo before reaching its breaking point [75]. The variant with 5% WBP addition showed the highest value of cohesiveness, while the control variant showed the lowest value. In the case of chewiness, a different correlation was noted: although the B5 variant still showed the highest value, it was the lowest for the variant with 10% WBP concentration. Similar results were obtained in a study of the effect of alternative proteins, including *Alphitobius diaperinus* powder, on the textural profile of bread [76]. Another study showed that TPA of bread is also influenced by the species of insect protein [77]. Using cricket flour instead of soy protein isolate significantly improved the protein profile and textural properties of the meat analogues [53,78]. However, 15% and 30% additions of mealworm powder to analogues made from plant proteins results in a reduction in their hardness, cohesiveness, springiness, and chewiness, while improving the degree of water binding [72]. These effects are probably caused by the weakening of molecular interactions in the soy protein network by mealworm proteins, which is similar to the interactions between insect and gluten proteins in the formation of cross-linked structures.

### 3.3. Sensory Evaluation

In the consumer evaluation of the sensory characteristics (color, aroma, firmness, juiciness, taste, aftertaste, and overall acceptability) of the burger analogues, no significant statistical differences were found (*p* < 0.05) for all tested variants. Despite the variation in WBP concentration, the sensory profiles remained consistent, indicating that the addition of WBP did not significantly alter the sensory attributes of the burger analogues. Figure 3 presents the results in graphical form. Regarding appearance, the 5% addition of WBP (B5) received the highest rating (5.75), surpassing the control (B0) at 5.63 and the 10% addition (B10) at 4.92. In terms of color, B5 achieved the greatest score (6.29), while both B0 and B10 scored equally at 5.33. Flavor scores were comparable across all treatments, with B10 obtaining the highest mark (5.96), followed by B5 (5.79) and B0 (5.75). Texture firmness was most preferred in B5 (5.63), trailed by B10 (5.04) and B0 (4.92). Juiciness scores were equivalent for B5 and B10 (5.46), while B0 scored 4.96. Taste was most favored in B10 (5.79), compared to B5 (5.75) and B0 (5.46). Aftertaste showed a preference for B5 (5.83) over B0 (5.42) and B10 (5.63). Finally, overall liking was greatest for B5 (5.88), followed by B10 (5.46) and B0 (5.25). The mean scores for all sensory attributes were around 5.0, indicating that the burger analogues were generally well accepted by the panelists. It can be observed that a moderate addition of WBP (5%) improve the visual appeal of the burgers, but a higher concentration (10%) detracts from it. The addition of WBP to the burger analogues also slightly improved the perception of the smell, texture, and taste of the product. Caparros Megido et al. [19] studied the effect of the addition of insect protein on the sensory aspects of beef burgers and plant-based burger. It was shown that the addition of insect protein to the beef burger lowered the overall product acceptability score, while the value was higher when added to the lentil burger analogue. This may indicate that the addition of insect protein mimics the taste of conventional meat. The results presented by Smetana et al. [79] showed that the insect burger had a significantly higher overall acceptability score compared to the plant-based burger available in most supermarkets. The insect burger was also rated higher for flavor and texture, while the plant-based alternative received higher scores for appearance. The sensory evaluation indicates that, while the addition of WBP can enhance certain sensory characteristics of the burger analogues, the optimal concentration for overall sensory appeal appears to be around 5%. Further investigations using a larger sample size or trained sensory panels could be conducted to explore these potential subtle differences. Moreover, different types and concentrations of spices could be studied to improve the sensory evaluation of the tested products.

The implementation of insect protein in food products is the subject of a number of scientific papers, focusing on the addition of this novel food to existing products in various forms [39,62,65,76]. Some of these studies do not respect legislation aspects, which may be varied in different countries. There are specific limits included in Commission Implementing Regulation (EU) 2023/58 of 5 January 2023 and amend the Implementing Regulation (EU) 2017/2470. The first one authorize the sale of frozen, paste, dried, and powder forms of *Alphitobius diaperinus* larvae (lesser mealworm) as a novel food, the other one establishing the addition of various species, forms, and concentrations of insect protein to products such as pasta, biscuits, oatmeal, and meat analogues, among others.

## 4. Conclusions

This research aims to show the effect of enriching plant-based meat analogues with insect protein from *Alphitobius diaperinus* on the physicochemical properties and sensory acceptability. The inclusion of Whole Buffalo Powder in the burger analogues leads to a reduction in pH (6.34–6.78) and a reduction in the protein content from 17.99% for B10 to 20.17% for B0, respectively. Partial substitution of soy protein with WBP in burger analogues alters the overall fat content and fatty acid composition of burger analogues, leading to an increase in total lipid content (31.00–35.26) and saturated fatty acids (11.28–21.28), while monounsaturated (5.24–8.31) and polyunsaturated fatty acids (8.74–11.41) decrease. Regarding the physical properties of the products obtained, significant differences were observed in color: with increasing WBP concentration the color values decreased. There was no significant effect of the addition of insect protein on the cooking efficiency and on the parameters of TPA. The results obtained after the sensory acceptance examination of the variants indicated that the variant with 5% WBP content was considered to be the best. With the results obtained, it is clear that edible insects can be successfully used as an additive in plant-based burger analogues or when producing a variety of plant-based meat alternatives, as their addition does not significantly affect the physicochemical attributes of the product, which was proven via sensory analysis. These findings can be helpful in the development of foods enriched with insect protein for use in countries where eating insects is unknown.

## Figures and Tables

**Figure 1 foods-13-01806-f001:**
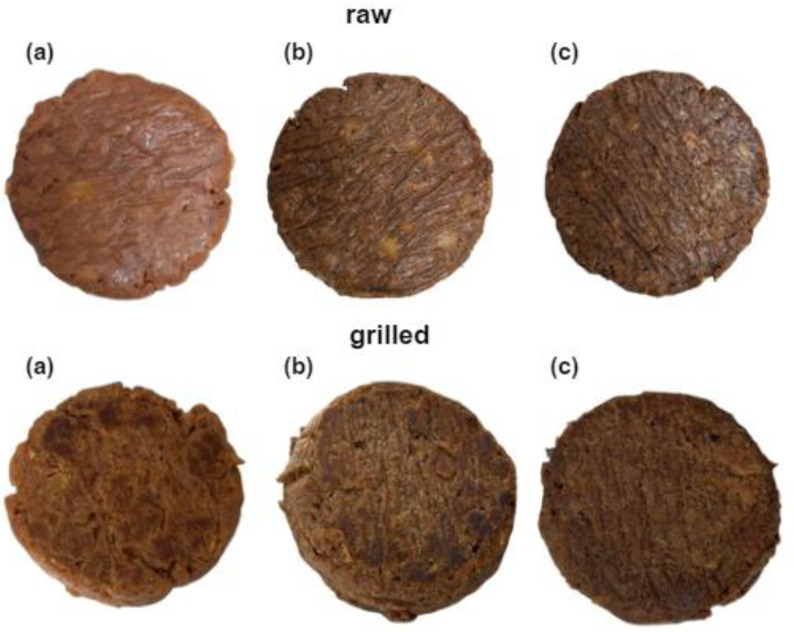
Variants of burger analogues with different WBP contents before and after being grilled: (**a**) B0, (**b**) B5, and (**c**) B10.

**Figure 2 foods-13-01806-f002:**
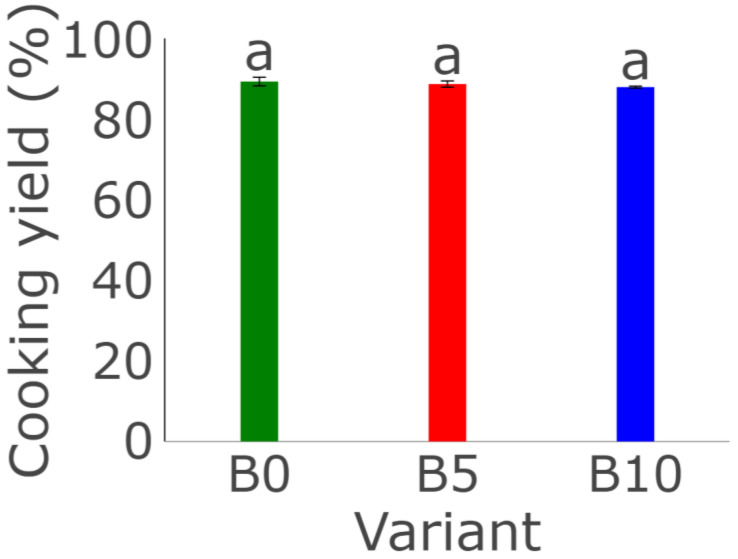
Effect of WBP content on cooking yield of burger analogues. The data represent the mean and the error bars represent the standard deviation (*n* = 3). The values indicated by different lowercase letters were significantly different (*p* ≤ 0.05).

**Figure 3 foods-13-01806-f003:**
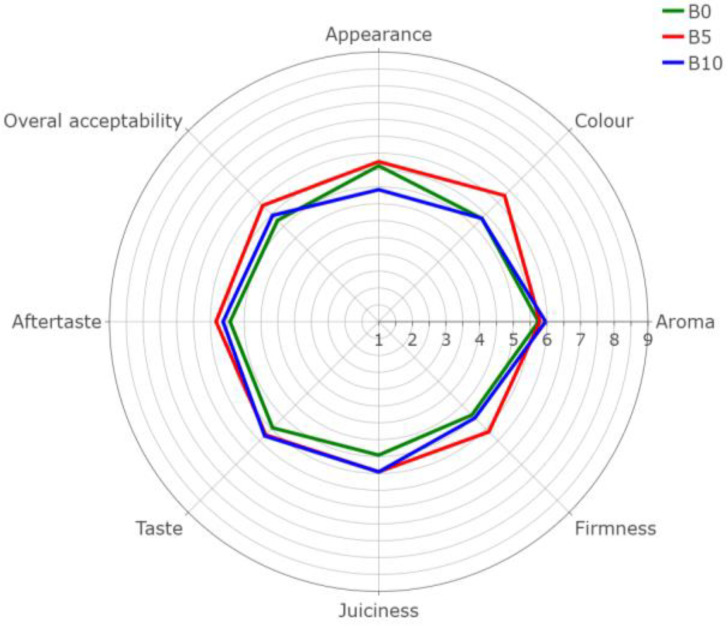
Spider plot of the sensory profile of burger analogues with the addition of WBP at different concentrations. For the identification of variants codes, refer to Table 1. The data represent the mean (*n* = 24).

**Table 1 foods-13-01806-t001:** Burger analogues variants composition [%]. B0—control, without added insect protein; B5—with 5% WBP content; and B10—with 10% WBP content.

Ingredients [%]	Burger Analogues		
	B0	B5	B10
Soy chop ^1^	13.6	6.8	6.8
Soy protein isolate GS5200 A ^2^	10	7.4	4.4
Fiber Textured Insect Protein	-	21.6	21.6
Whole Buffalo Powder	-	5	10
Sodium Alginate FD 901 AR ^3^	1.4	1.4	1.4
Transglutaminase ACTIVA WM ^4^	1.5	1.5	1.5
Refined sunflower oil ^5^	7	7	7
Beetroot juice BIO ^6^	2	2	2
Spices:			
- Salt ^7^	- 1.3	- 1.3	- 1.3
- Pepper ^8^	- 0.5	- 0.5	- 0.5
- Smoked paprika ^9^	- 1	- 1	- 1
- Spicy paprika ^10^	- 0.2	- 0.2	- 0.2
- Garlic ^10^	- 0.5	- 0.5	- 0.5
- Cumin ^8^	- 0.5	- 0.5	- 0.5
- Nutmeg ^8^	- 0.2	- 0.2	- 0.2
Virgin coconut oil ^11^	7	7	7
Water	53.3	46.3	44.3

(^1^) Sante Sp. z o.o, Warsaw, Poland; (^2^) Novichem Sp. z o.o., Chorzów, Poland; (^3^) Danisco GRINDSTED^®^, Grindsted, Denmark; (^4^) Ajinomoto Foods Europe SAS, Paris, France; (^5^) EOL (Edible Oils Limited) Polska Sp. z o.o, Szamotuły, Poland; (^6^) Naura, Białystok, Poland; (^7^) P.P.H. “STANLAB” s.j., Lublin, Poland; (^8^) McCormick Polska S.A., Stefanowo, Poland; (^9^) Prymat Sp. z o.o, Jastrzebie Zdroj, Poland; (^10^) ŻUK-POL Sp. z o.o, Wrocław, Poland; (^11^) Żywność Ekologiczna Bio Food Sp. z o.o., Ciechocin, Poland.

**Table 2 foods-13-01806-t002:** Chemical properties (pH and protein contents) of burger analogues with different WBP concentrations (0–10%).

Variant	pH [-]	Protein Content [%]
B0	6.78 ^a^ ± 0.03	20.17 ^a^ ± 0.79
B5	6.37 ^b^ ± 0.02	18.41 ^ab^ ± 1.08
B10	6.34 ^b^ ± 0.03	17.99 ^b^ ± 0.24

The data are expressed as mean ± standard deviation (n = 3). The means in the columns with different superscript letters exhibited significant differences (*p* ≤ 0.05).

**Table 3 foods-13-01806-t003:** Effect of WBP addition at different concentrations on burger analogues’ fatty acid profile.

Variant	Lipids Total [% *w*/*w*]	SFA [g/100 g]	MUFA [g/100 g]	PUFA [g/100 g]
B0	31.00 ^c^ ± 0.20	11.28 ^c^ ± 0.013	8.31 ^a^ ± 0.003	11.41 ^a^ ± 0.016
B5	33.12 ^b^ ± 0.17	14.69 ^b^ ± 0.038	7.78 ^b^ ± 0.004	10.67 ^b^ ± 0.002
B10	35.26 ^a^ ± 0.14	21.28 ^a^ ± 0.004	5.24 ^c^ ± 0.003	8.74 ^c^ ± 0.003

The data are expressed as mean ± standard deviation (*n* = 3). The means in the columns with different superscript letters exhibited significant differences (*p* ≤ 0.05). SFAs: saturated FAs; MUFAs: monounsaturated FAs; PUFAs: polyunsaturated FAs.

**Table 4 foods-13-01806-t004:** Influence of WBP addition at different concentrations on burger analogue color.

Variant	Raw			Grilled		
	L*	a*	b*	L*	a*	b*
B0	44.44 ^a^ ± 0.28	12.15 ^a^ ± 0.16	16.21 ^a^ ± 0.13	35.65 ^a^ ± 0.16	15.63 ^a^ ± 0.16	15.63 ^a^ ± 0.16
B5	42.43 ^b^ ± 0.20	8.30 ^b^ ± 0.27	13.50 ^b^ ± 0.27	32.78 ^b^ ± 0.36	9.65 ^b^ ± 0.45	13.97 ^b^ ± 0.14
B10	39.41 ^c^ ± 0.12	7.04 ^c^ ± 0.12	11.41 ^c^ ± 0.40	28.53 ^c^ ± 0.38	8.96 ^b^ ± 0.36	10.67 ^c^ ± 0.21

The data are expressed as mean ± standard deviation (*n* = 3). The means in the columns with different superscript letters exhibited significant differences (*p* ≤ 0.05).

**Table 5 foods-13-01806-t005:** Influence of WBP addition at different concentrations on texture profile analysis of burger analogues.

Variant	Hardness [N]	Cohesiveness [-]	Springiness [-]	Gumminess [N]	Chewiness [N × mm]
B0	9.69 ^a^ ± 1.04	0.40 ^b^ ± 0.04	0.69 ^a^ ± 0.06	3.88 ^a^ ± 0.15	2.67 ^a^ ± 0.27
B5	7.74 ^a^ ± 0.85	0.52 ^a^ ± 0.04	0.57 ^ab^ ± 0.14	4.00 ^a^ ± 0.60	2.29 ^a^ ± 0.12
B10	7.85 ^a^ ± 1.01	0.41 ^b^ ± 0.06	0.50 ^b^ ± 0.07	3.29 ^a^ ± 0.76	1.61 ^b^ ± 0.38

The data are expressed as mean ± standard deviation (*n* = 3). The means in the columns with different superscript letters exhibited significant differences (*p* ≤ 0.05).

## Data Availability

The original contributions presented in the study are included in the article, further inquiries can be directed to the corresponding author.

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
