# Peer review of "Insect Protein as a Component of Meat Analogue Burger"

_foods, 2024, doi:10.3390/foods13121806_

Round 1
Reviewer 1 Report
Comments and Suggestions for Authors
Insect is an ideal bio-resource for food and feed industry. The manuscript 'Insect protein as a component of meat analogue burger' discloses the effect of insect-based ingredients (insect protein from Alphitobius diaperinus) on the nutritional values of meat analogue burger. The well-written manuscript focuses on an interesting and topical subject and is of considerable practical importance; however, some content need to be revised.
The title should be changed to meat slices for hamburgers, not ‘meat analogue burger’
LINE 57 Does this insect Alphitobius diaperinus food meet the needs of most people from a dietary pagoda perspective?
LINE 113 Could authors provide more information on Alphitobius diaperinus, because the nutritional value of this insect and products has been researched for many years in some countries outside Europe.
LINE 129 In variants preparation, what is the effect of added oils, flavourings and other ingredients on chemical properties adn physical properties?
LINE 244 Could authors provide more information on the role of Alphitobius diaperinus in future food industry in Discussion?
LINE 49-54 Please provide the citation.
LINE 123 Have you analyzed the chemical properties adn physical properties of the insect protein from Alphitobius diaperinus? Supplement the experiment data would be better.
LINE 209 which CIE-lab colour space?
LINE 307 Please replace with a clearer figure 2.
Reviewer 2 Report
Comments and Suggestions for Authors
The article is interesting, but the authors should improve several features. The experimental work is good, but the experimental design needs to be more scientifically sound.
- The authors could have evaluated others post-processing, such as microwave cooking, oil-frying, etc.
- What are the criteria for selecting the following: B5 with 5% WBP content and B10 with 10% WBP content?
- The authors could have assessed the effect of grilled temperature or other operative conditions.
- Figure 2 needs to be sharp and straightforward.
- The authors could have used thickness, such as an operative condition, to evaluate the quality of the samples' characteristics.
- How was the temperature at 72°C measured for the sample's geometric center? What kind of devices were used?
- What are the criteria for selecting for hardness [N], cohesiveness [-], and chewiness [N x mm] of samples?
- The conclusion needs more numerical values, percentages, comparisons, increasing/decreasing, trends, etc.
Reviewer 3 Report
Comments and Suggestions for Authors
The study developed soy protein-based burger enriched with insect protein, which is an interesting topic in recent years. The experiments were well-designed and the results were well-discussed, which reflects the expertise of the authors. The introduction gives a detailed and convincing argument about the importance of this study. Here are some tips:
1. The authors used “Alphitobium diaperinus” and “whole buffalo powder” in the manuscript, what’s the relation between them, please clarify.
2. What’s the usage of fiber textured insect protein? And the authors added the fiber textured insect protein at a 21.6% portion. Actually, the burger properties were not affected by the whole buffalo powder, but also fiber textured insect protein. How do you explain it?
3. From the nutrient aspect, the protein content decreased and lipid content increased with the insect powder addition. Does that means the nutrient profile was not improved by the insect powder addition?
Reviewer 4 Report
Comments and Suggestions for Authors
"The objective of this manuscript is very interesting, especially considering the market opportunities in Western countries. The research design is clear and well-structured but could be further improved. The literature review is thorough. I would recommend some improvements to make the article more comprehensive and readable. From my perspective, I suggest including a brief analysis of several systematic reviews conducted in recent years, particularly those focused on Europe, to enhance the market framework. It would be beneficial to explicitly state the research questions. There are no issues with the methodology or the presentation of results. I would suggest explicitly discussing the study's limitations in the conclusion."
Reviewer 5 Report
Comments and Suggestions for Authors
The manuscript investigates the inclusion of insect protein (Alphitobius diaperinus) in meat analog burgers based on soy protein. Three formulations were developed: control (B0), with 5% (B5) and 10% (B10) of insect protein. Physicochemical properties, fatty acid profile, cooking performance, texture profile analysis and sensory acceptability were evaluated.
I found some points that must be justified to support the relevance of this research:
1. The addition of insect protein resulted in a significant decrease in the total protein content in the B5 and B10 variants compared to the control. This may be a negative aspect from a nutritional perspective and could have been addressed better, perhaps by adjusting other components of the formula to compensate for this reduction.
2. Although the addition of insect protein increases the content of total lipids and saturated fatty acids (SFA), it also decreases monounsaturated fatty acids (MUFA) and polyunsaturated fatty acids (PUFA), which can be a negative point for health perception of the product.
3. Although there were no significant differences in sensory evaluation, the variant with 10% insect protein received lower scores compared to the 5% variant. This suggests that a higher concentration of insect protein may negatively affect the sensory perception of the product.
4. The discussion sometimes lacks a clear connection to the results presented, and there is redundancy in some sections that could be reduced to improve clarity and conciseness.
5. Alphitobius diaperinus can cause hypersensitive reactions in people with allergies to crustaceans and dust mites. Why does the study not adequately address the allergenic potential of insect protein?, a critical aspect for consumer acceptance and food safety.
About the document
On lines 15 to 17 there are acronyms that need to be defined (MUFA, PUFA)
Round 2
Reviewer 5 Report
Comments and Suggestions for Authors
The observations later discussed in the manuscript.